# Polish Translation and Validation of the Voice Handicap Index (VHI-30)

**DOI:** 10.3390/ijerph191710738

**Published:** 2022-08-29

**Authors:** Beata Miaśkiewicz, Elżbieta Gos, Małgorzata Dębińska, Aleksandra Panasiewicz-Wosik, Dorota Kapustka, Katarzyna Nikiel, Elżbieta Włodarczyk, Anna Domeracka-Kołodziej, Paulina Krasnodębska, Agata Szkiełkowska

**Affiliations:** 1Audiology and Phoniatric Clinic, World Hearing Center, Institute of Physiology and Pathology of Hearing, 05-830 Warsaw, Poland; 2Department of Teleaudiology and Screening, World Hearing Center, Institute of Physiology and Pathology of Hearing, 05-830 Warsaw, Poland; 3Rehabilitation Clinic, World Hearing Centre, Institute of Physiology and Pathology of Hearing, 05-830 Warsaw, Poland

**Keywords:** voice handicap index, quality of life, reliability, validity, questionnaire

## Abstract

Traditional voice evaluations, including imaging techniques, auditory-perceptual ratings, and acoustic and aerodynamic analyses, fail to assess the global handicap that a patient experiences as a result of a voice disorder. The Voice Handicap Index (VHI) is currently one of the most widely used and psychometrically robust instruments for measuring voice disability. The aim of the study is to translate and validate a Polish version of the VHI. The original English-language version of VHI-30 was translated into Polish. We enrolled 188 subjects—123 patients (91 women and 32 men) with voice disorders and 65 controls (53 women and 12 men) without voice disorders. Results obtained by the patients were significantly higher than those obtained by the controls on the Emotional subscale (*U* = 519.0; *p* < 0.001), Functional (*U* = 829.0; *p* < 0.001), Physical (*U* = 331.0; *p* < 0.001), and the global score (*U* = 390.0; *p* < 0.001). There were statistically significant negative correlations between maximum phonation time and global score (*rho* = −0.31; *p* < 0.01) as well as all three subscales. Shimmer and Smoothed Amplitude Perturbation Quotient were correlated positively with the global score (*rho* = 0.22; *p* < 0.05; *rho* = 0.25; *p* < 0.01, respectively) and with all three subscales. There were also statistically significant correlations between VHI scores and auditory perceptual evaluation. In the patient group, there was excellent internal consistency (α = 0.97) and strong test–retest reliability (intraclass correlation = 0.94). The cut-off value equal to 17 points was estimated. The Polish VHI showed excellent internal consistency, good test–retest reproducibility, and clinical validity. It is a useful tool for evaluating the voice disability perceived by a patient.

## 1. Introduction

Voice disorders can affect patients in many different ways and may have an impact on their quality of life, creating numerous psychological, social, physical, and occupational implications [1]. Traditional evaluations, including visual-perceptual measures or objective acoustic and aerodynamic analyses, fail to assess the global handicap that a patient experiences as a result of a voice disorder [1]. In recent decades, particular importance has been attached to the assessment by patients of the subjective impact of voice disorders on the quality of life. Since the same voice disorder may result in different patient handicaps, for clinical application, standardized tools should be administered to evaluate the handicap experienced by a given patient [2]. Handicap can be thought of as a reduction of the quality of life.

Researchers have shown the importance of including subjective self-measurements performed by the patient in standard voice evaluation [3]. The European Laryngological Society recommends using self-reported questionnaires as a part of a multidimensional voice analysis protocol [4]. Instruments measuring quality of life in dysphonic patients include the Voice Handicap Index (VHI) [5], the Voice-Related Quality of Life (V-RQOL) [6], Voice Activity and Participation Profile [7], and Voice Symptom Scale [8]. In particular, the VHI is reported to meet the criteria for reliability, validity, and availability of normative data by the Agency for Health Care Research and Quality [1]. It has proved to be the most versatile and easiest-to-score instrument, which provides the most relevant item information among the commonly used voice-related quality of life instruments [9]. The VHI is a self-administered tool that can be used with patients presenting a wide range of voice disorders, not only those originally related to the larynx, but also such as Parkinson disease, multiple sclerosis, gastroesophageal reflux, as well as after treatment of benign and malignant vocal fold lesions [2,3].

The VHI consists of 30 items with five response levels, scored 0–4. These items are equally distributed into three subscales: functional (VHI-F), physical (VHI-P), and emotional (VHI-E). The functional subscale gauges the ability to communicate in various settings, the physical addresses the patient’s perceptions of laryngeal discomfort and voice output characteristics, and the emotional subscale measures emotional aspects of voice production [5,10]. Since its development and first validation in the USA, the VHI has been translated and validated into several languages, including Serbian, Brazilian Portuguese, Chinese, German, Dutch, Spanish, Italian, Greek, Arabic, Croatian, Persian, Latvian, and Slovak [2,3,9,11,12,13,14,15,16,17,18,19,20]. International comparisons that focused on identifying the differences between translated versions due to linguistic or cultural reasons proved the equivalence of some VHI versions and the original VHI version [21]. The standard procedure for translating into a non-US-English language, taking into account language and cultural adaptation, include several parallel translations, review of translations to get a consensus version, back-translation and comparison with the original version, and finally, pilot testing of the translation [10,21].

The importance of developing patient-based assessment instruments for use with objective and perceptual measures for the evaluation of voice disorders and treatment efficacy is cited [3,4]. However, some works have shown large discrepancy between the VHI and typical measurements obtained in the voice laboratory [13,22,23]. Although there is no strong relationship between the VHI and acoustic and aerodynamic parameters, it has been used clinically to assess patient-perceived severity of voice disorder and treatment outcomes [22,23].

Currently, there is one adapted, statistically valid, and reliable instrument for measuring voice disorder handicap in Poland (V-RQOL) [24]. A Polish version of the VHI was provided in 2004 by Pruszewicz et al. [25]. However, the authors themselves acknowledged that the tool had been modified by them. The content of some of the items was changed without justification, and the order of the items was changed too. The translation procedure was not described and no evidence was given that the modified tool maintained conceptual equivalence with the original. Since the VHI is the most widely used and with the greatest psychometric potential, adaptation and validation of the VHI for Polish, accordingly to standard procedures, would enable within language and cross-language comparisons of voice disorder severity and comparability of treatment outcomes.

The aim of this study is to translate and validate a new Polish version of VHI, which we will refer to as VHI-POL.

## 2. Materials and Methods

### 2.1. Translation from English to Polish

The original English-language version of the VHI was translated into two separate Polish texts by two qualified professional translators. Then, five experts (four phoniatricians and an ENT specialist), all fluent in both Polish and English languages, discussed the translations of each item and chose the more appropriate version of the two. Experts were in full agreement on 10 items, on the remaining 20 they debated which version to choose until they reached a consensus. The whole tool was then back-translated into English by a professional Polish native-speaking translator who was unaware of the original questionnaire. The experts compared this version with the original English-language version and consulted together. They concluded that the two versions did not differ substantially in any respect, and at that point it was decided to submit this preliminary version of the VHI to a pilot study in the target population, that is, patients seeking help because of their voice disorders.

A pilot study of the preliminary Polish version of the VHI was conducted on 30 randomly chosen adult patients with voice disorders, who had been referred to our clinic—the tertiary referral center. There were 16 women, 9 men, 5 subjects did not provide gender information. They were aged between 34 and 81 years (mean age 55.5; *SD* = 12.6).

This pilot testing was intended to check the ease of understanding, relevance, acceptability, and feasibility of the new questionnaire [26]. After the patients had completed the preliminary version of the tool, they were asked about their experience, and all of them reported that the items were understandable. Some 93% of the patients reported that the content of the items was relevant to their voice problems. The acceptability of the questionnaire was high: 90% of the patients said that they would be willing to fill in this tool during a medical consultation and 97% percent found the questionnaire practical and said they were able to complete it. The results of the pilot study were discussed by the expert panel, which led to the next phase of the investigation—establishing the psychometric properties of the final VHI-30 Polish version (see Appendix A, Voice Handicap Index Polish Version (VHI-POL)) in a larger clinical sample.

### 2.2. Setting

Patients were recruited to the investigation between November 2018 and September 2021.

All participants gave their written informed consent for participating in the study. The study protocol and the informed consent form were approved by the Institutional Ethics Committee (Institute of Physiology and Pathology of Hearing; approval number KB.IFPS: 22/2019).

All participants were asked to complete the Polish VHI (VHI-POL). Following completion of the questionnaire, participants in the study group underwent a comprehensive voice evaluation, including a complete head and neck examination, laryngo-video-stroboscopy (LVS), measurement of maximum phonation time, and objective and auditory-perceptual voice evaluations. At the next visit (about 1–3 months later), the patients were asked to fill in the VHI-POL again and answer one additional question about the self-reported change in their voice status compared to the first administration. The participants did not receive any therapeutic intervention during that time.

The patient’s evaluation protocol used in the Audiology and Phoniatric Clinic is based on guidelines elaborated by the Committee on Phoniatrics of the European Laryngological Society [4]. Laryngo-video-stroboscopy was performed using a 70° rigid laryngoscope (EndoStrob DX Xion 327, GmbH, Xion, Berlin, Germany). The maximum phonation time (MPT) was determined by measuring the duration of sustained production of the vowel “a”, and the maximum value of the three productions was taken as the representative MPT [4,27].

Objective acoustic voice analysis was performed with a Computerized Speech Lab (CSL) 4500 external module from Kay Elemetrics Corporation (Lincoln Park, NJ, USA). All voices were recorded with an ECM 800 microphone (Behringer, Behringer Holdings (Pte) Ltd., Singapore), positioned approximately 15 cm from the mouth at an angle of 45° to reduce airflow effects. Analysis of a voice sample recorded at a sample rate of 25 kHz was conducted using the Multidimensional Voice Program software (MDVP 5105 version 2.7.0, Kay Elemetrics Corporation, Lincoln Park, NJ, USA). Three samples of the sustained vowel “a” in modal voice were used for analysis; only the middle portion of the uttered vowel was used (min. 0.6 s), avoiding onset and offset effects [28]. The following acoustic parameters were calculated: average fundamental frequency (F0), frequency variation (% Jitter; Relative Average Perturbation, RAP; Pitch Perturbation Quotient, PPQ; Smoothed Pitch Perturbation Quotient, sPPQ; Fundamental Frequency Coefficient Variation, vF0), amplitude variation (% Shimmer; Amplitude Perturbation Quotient, APQ; Smoothed Amplitude Perturbation Quotient, sAPQ; Peak-to-Peak Amplitude Coefficient of Variation, vAm), and noise-related parameters (Noise to Harmonic Ratio, NHR; Soft Phonation Index, SPI).

Auditory- perceptual evaluation of the patients’ voices was carried out on the GRBAS scale [29] in which the clinician estimates the Grade of hoarseness (G), Roughness (R), Breathiness (B), Asthenia (A), and Strain in the voice (S) on a scale from 0 to 3 (0, normal; 1, mild; 2, moderate; 3, severe). Ratings based on the patient’s sustained phonation and a short speech sample were made by the senior author (experienced phoniatrician and ENT specialist) upon each presentation to the clinic. Then, the retrospectively performed blinded evaluation of the recorded voice samples was carried out by the same researcher.

Initially, 196 subjects were recruited (128 patients with a voice disorder and 68 controls). Since 5 patients and 3 controls did not fill in the VHI-30 completely, they were excluded from further analysis. Finally, there were 188 subjects—123 in the study group and 65 in the control group.

### 2.3. Participants

The inclusion criteria were: age over 18 years, voice disorder diagnosed and confirmed by an ENT doctor or phoniatrician, Polish-language fluency, and ability to fill in the questionnaire. The control group consisted of healthy non-dysphonic adult family members of patients, or friends of health care professionals, who did not report a history of voice problems. The exclusion criteria for both patients and controls were: cognitive disability, neurological disturbance, or inability to fill in the voice quality questionnaire without assistance.

We recruited 123 patients (91 women and 32 men) with voice disorders and 65 controls (53 women and 12 men) without voice disorders.

The patients were aged between 18 and 77 years (*M* = 50.5; *SD* = 13.6), and the controls were aged between 24 and 63 years (*M* = 41.9; *SD* = 11.7). Professional voice users (teachers) made up 63% of the study group and 48% of the controls. Following laryngeal examination, the patients’ voice disorders were diagnosed: there were 40 after previous vocal fold (VF) surgery, 39 with functional dysphonia (primary muscle tension dysphonia), 24 with organic benign vocal fold (VF) lesions, and 19 with unilateral vocal fold paralysis (UVFP).

### 2.4. Psychometric and Statistical Analyses

Results of the VHI-POL were investigated by descriptive statistics for all items, subscales, and global scores. Discriminative validity was assessed by comparison of the VHI-POL scores between the study and control groups. The predefined hypothesis was that the patients would have higher VHI-POL scores than the controls and this was checked with a Mann–Whitney test. In addition, we checked whether patients with various diagnoses differed in terms of VHI-POL scores. It was hypothesized that patients with UVFP would obtain higher scores than those with other diagnoses, and this was checked with a Kruskal–Wallis test. Criterion validity was evaluated by rho-Spearman bivariate correlations. The VHI-POL was predicted to correlate negatively with the MPT, positively with MDVP parameters, and positively with GRBAS assessment. Criteria provided by Cohen [30] were used to interpret the strength of correlation: weak (<0.3), moderate (0.3–0.5), or strong (>0.5). Internal consistency was measured as Cronbach’s α coefficient. According to the criterion given by Nunnally and Bernstein [31], internal consistency was considered good when α was above 0.70. To determine the reproducibility of the VHI-POL, intraclass correlation (ICC) was used, with a positive rating over 0.70 [32]. Reproducibility was assessed only in that subgroup of the patients who reported stable health between test and retest. Limits of agreement were also calculated for this group according to Bland and Altman [33], with 95% of scores expected to be within the identified agreement limits. Receiver Operating Characteristic (ROC) analysis was conducted for estimating the cut-off point for VHI-30. The cut-off is a value that best discriminates between results of a test classified as positive (i.e., with presence of disorder) or negative (absence of disorder). The ROC method combines information on sensitivity (i.e., true positive rate) and specificity (i.e., true negative rate) [26]. The area under the ROC curve (AUC) should be at least 0.7 to be considered acceptable. Statistical analysis was conducted with IBM SPSS Statistics v.24 (IBM Corp., Armonk, NY, USA), and a *p*-level < 0.05 was considered statistically significant.

## 3. Results

### 3.1. VHI-POL Items, Subscales, and Global Score

Descriptive statistics (minimum, maximum, mean, and standard deviation) for all items, subscales, and global scores are given in Table 1. For the study group, scores were the highest on item 23 (upset about voice problem), item 4 (variety in the sound of voice throughout the day), and item 14 (strain while producing voice). Statistically significant differences between the study and control groups were found for all items (*p* < 0.001), with patients scoring higher than controls.

Statistically significant differences between the study and control groups were found on the Emotional subscale (*U* = 519.0; *p* < 0.001), Functional (*U* = 829.0; *p* < 0.001), Physical (*U* = 331.0; *p* < 0.001), and the global score (*U* = 390.0; *p* < 0.001). Results obtained by the patients were significantly higher than those obtained by the controls.

The four groups of patients with various diagnoses were compared in terms of VHI-POL scores. The type of diagnosis was found to be a significant factor for all three subscales: VHI-E: χ2 = 10.90, *p* = 0.012; VHI-F: χ2 = 15.75, *p* = 0.001; VHI-P: χ2 = 12.69; *p* = 0.005. Additionally, the global score was significantly different between the four groups of patients (χ2 = 13.71; *p* = 0.003). The patients with UVFP obtained the highest global scores (*M* = 68.75; *SD* = 27.06), while scores obtained by other patient groups were lower: functional dysphonia (*M* = 50.00; *SD* = 23.29; *U* = 229.5; *p* = 0.010), organic VF lesions (*M* = 42.63; *SD* = 22.10; *U* = 111.5; *p* = 0.020), and sequelae after previous VF surgery (*M* = 42.30; *SD* = 24.81; *U* = 193.5; *p* = 0.001). The means scores are shown in Figure 1. 

### 3.2. Criterion Validity

There were statistically significant negative correlations between MPT and emotional (*rho* = −0.26; *p* < 0.01), functional (*rho* = −0.27; *p* < 0.01), and physical subscales (*rho* = −0.36; *p* < 0.001), as well as for the global score (*rho* = −0.31; *p* < 0.01). The higher the VHI-POL score, the shorter the MPT.

Statistically significant correlations were also found between MDVP parameters and VHI-30 scores (Table 2). Two parameters, Shim and APQ, were correlated positively with all three subscales as well as the global score.

There were also statistically significant correlations between VHI-POL scores and the GRBAS evaluation (Table 3). The correlations were positive, meaning that higher severity of voice complaints was accompanied by higher scores in VHI-POL.

### 3.3. Internal Consistency 

Correlations between the subscales were strong: VHI-E and VHI-F, *rho* = 0.89; VHI-E and VHI-P, *rho* = 0.82; VHI-F and VHI-P, *rho* = 0.80. All subscales were correlated with the global score at a level above 0.90. Cronbach’s alpha values (in the study group) were as follows: VHI-E, α = 0.94; VHI-F, α = 0.94; VHI-P, α = 0.90; global score, α = 0.97.

### 3.4. Reproducibility and Agreement 

The reproducibility of the VHI-POL measured by ICCs was assessed only in the subgroup of stable patients, i.e., those who perceived no change in their voice status at the retest compared to the first test (n = 69). For VHI-E, ICC was 0.93, for VHI-F it was 0.93, for VHI-P it was 0.92, and for VHI-POL global it was 0.94. Table 4 shows agreement between test and retest. 

### 3.5. Cut-Off Value for VHI-POL Global Score

In the ROC analysis, the distinction was made between the patients with voice disorders and the controls. The result obtained for the VHI-POL global score is shown in Figure 2.

## 4. Discussion

The Voice Handicap Index is a 30-item questionnaire introduced by Jacobson et al. as an instrument to quantify the psychosocial consequences of a voice disorder, either in terms of voice outcome or voice-related quality of life. Jacobson and colleagues showed that the VHI has good internal consistency, test–retest reliability, and correlation with clinical findings of voice-related severity [5]. In 2002, the Agency for Health Care Research and Quality reported that the VHI met their criteria for reliability, validity, and availability of normative data [34]. Adaptation and validation of the VHI for a specific language provides within-language and cross-language comparisons of voice disorder severity and of the progress of treatment.

In the present study, we assessed the psychometric properties of the VHI-POL in measuring the effect of dysphonia on the patient’s everyday life.

The VHI-POL showed clinical validity for the overall VHI score, as well as for scores on its physical, functional, and emotional subscales. In our research, the mean total score as well as the three subscale scores were significantly higher in the study group than in the controls, demonstrating that the VHI can discriminate individuals suffering voice impairment from those who do not. Our findings are comparable with those of other researchers [2,3,9,13,14,15,16,17,18,19,20,35]. As in other studies, our patient group also reported the highest VHI-30 scores on the physical subscale, whereas the emotional and functional subscales were less affected [2,14,16,19]. This shows that physical symptoms are the most prominent self-reported voice problem, a result consistent with other findings [3,17,19,21,36,37]. Trinite et al. and Bonetti et al. explain it by the fact that people tend to associate physical symptoms of voice disorders with how they perceive the sound of their voice, whereas functional limitations and changes in emotional condition are related to the efficiency of interpersonal communication and experience of the voice problem [17,19]. Comparison of the four patient groups, each with a different type of pathology, showed statistically significant differences in the global score as well as for the three subscales. As found by other authors, we obtained the highest total score in patients with UVFP [9,14,15,18,20,21], whereas in the other subgroups, there were no statistical differences [2,17,19].

There was a significant negative correlation between the VHI-POL and MPT for total as well as for subscale scores. Similar findings were found in other studies [38,39]. Schindler et al. reported significant correlations between MPT and the functional and physical subscales, while Dehqan et al., only for the physical domain in patients with functional dysphonia [38,39]. There was a high and significant correlation between the physical subscale and MPT in patients with UVFP [39]. These results are similar to other findings and indicate that reduced respiratory capacity and decreased MPT have a negative impact on vocal efficiency and voice quality and thus on the VHI score [27,37,40].

Objective acoustic measures allow one to assess voice disturbances related to the mass and tension of the vocal folds, as well as other biomechanical characteristics. In contrast to some studies that have indicated a large discrepancy between laboratory measurements and the VHI, we found significant correlations between VHI-POL scores and acoustic MDVP measures. In particular, the two amplitude parameters Shim and APQ correlated positively with all three subscales and the total score. Some authors consider amplitude variations as the most significant factors in determining the severity of phonation disorders [28]. We found that the highest correlations were observed for the physical subscale encompassing all studied objective acoustic parameters except F0, NHR, and SPI. The lowest correlation was found for the emotional subscale. This result confirms the overarching importance of the physical subscale. Karlsen et al. showed that the VHI score correlated with acoustic measures, particularly Jitter, Shimmer, and NHR, with about 10% variance [40]. Most previous studies have used the Pearson test to assess the correlation between acoustic measures and VHI scores [22,23,38,41]. Hsiung et al. found one fairly significant correlation between the functional subscale and NHR [22]. Woisard et al. found a fair correlation between the voice frequency range and the physical subscale score, and between acoustic measures and single items of the VHI [23]. Wheeler et al. reported a significant correlation between Shimmer, Jitter, and NHR and most VHI items, although they did not see a correlation between acoustic measures and overall VHI [41]. Schindler et al. suggested that in patients with a similar origin of dysphonia, a correlation between acoustic voice parameters and VHI might exist. They found weaker correlations in the whole group of dysphonic patients than in subgroups which had different origins of voice disorder. In patients with UVFP, the Jitter value correlated with the functional subscale, whereas in patients with vocal nodules, Jitter, Shimmer, and NHR correlated with the VHI functional domain [38]. Dehqan et al. showed strong and significant correlation between the Jitter, Shimmer, and HNR values and the physical scale for patients with UVFP, benign vocal fold lesions, whereas the correlation was relatively weak in functional dysphonia group [39].

Similarly to Fujiki et al. [42], the total score and subscores in the present study were significantly correlated with all GRBAS ratings. However, the rho values were in the weak and moderate range for all correlations (rho ranging from 0.24 to 0.36). For the emotional and functional scales, the strongest correlations were observed for the GRBAS rating of Asthenia (respectively 0.34 and 0.36); for the physical scale, the strongest correlations were for the Grade of hoarseness and Roughness (respectively 0.35 and 0.36). Fujiki et al. found a significant correlation between GRBAS ratings and the functional subscale and observed a stronger association between the emotional subscale and the Strain domain [35]. Similar findings were reported by Bonetti et al. who observed significant correlation with the overall VHI score and functional subscale with the degree of dysphonia perceived by the listeners, suggesting that the greater self-reported handicap is accompanied by more audible deviations in the voice quality [17]. Overall, these findings generally suggest that the total VHI-POL score as well as the sub-scores correlate with the self-reported voice quality.

The internal consistency of the VHI-POL appeared high, with Cronbach’s alpha coefficient values of 0.97 for the total score, and 0.94, 0.94, and 0.90 for the emotional, functional, and physical subscales, respectively. We also found strong correlations between subscales (*rho* ≥ 0.80). These findings are in agreement with the original study by Jacobson et al. and with results of similar research in many other languages [2,5,13,14,16,17,18,19,20]. Behlau et al. found good internal consistency with Cronbach’s alpha of 0.888 for the total score but two of the domains (functional and emotional) had strong internal consistency as was reported in the Hebrew and Greek versions of the VHI [3,15,35].

The Polish version of the VHI showed excellent test–retest reliability on both the global scale (0.94) and on the three subscales: emotional (0.93), functional (0.93), and physical (0.92). These findings showed a high level of reproducibility and are similar to the results of the original study by Jacobson et al. [5], and in those reported by other studies [2,3,9,13,14,15,16,17,18,19,20].

A cut-off point for the VHI-30 global score was established to be 17 points. This value distinguishes the subjects with and without voice disorders. The sensitivity and specificity were above 80%, indicating good diagnostic accuracy. Our results are similar to those reported by Ng et al. and Behlau et al.—cut-offs of 18.5 and 19 points, respectively [43,44]. However, Moradi et al. indicated cut-off as 14.5 points, which is more “restrictive” [45]. Thus, the question of how many points on the VHI at least is required to suspect a voice disorder remains an open question. In this respect, there may be cross-cultural differences and different cut-off points are appropriate for patients from different countries. Tafiadis et al. emphasized that the use of cut-off points is valuable in voice assessment, especially in primary health care services, but also is important for clinical decisions made by a voice clinician. However, it must be used in conjunction with results of other voice assessments [46].

The valid and reliable Polish version of the VHI will allow specialists to make more precise diagnosis and treatment strategies of Polish-speaking individuals with voice disorders and with updating the questionnaire will enable a reliable assessment as a scientific tool.

## 5. Conclusions

The VHI-POL showed excellent internal consistency, good test–retest reproducibility, and clinical validity. The outcomes of the present study confirmed that the VHI-POL is a useful tool for self-reported evaluation of voice patients. The questionnaire provides an additional piece of information in diagnosing voice disorders and can be used to clearly distinguish dysphonic patients from those without voice impairment.

## Figures and Tables

**Figure 1 ijerph-19-10738-f001:**
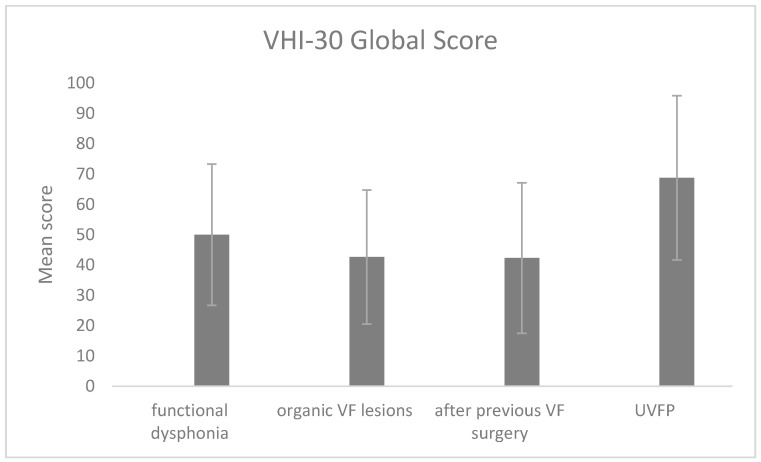
Mean VHI-POL global score depending on diagnosis. The error bars represent standard deviations. VF, vocal fold; UVFP, unilateral vocal fold paralysis.

**Figure 2 ijerph-19-10738-f002:**
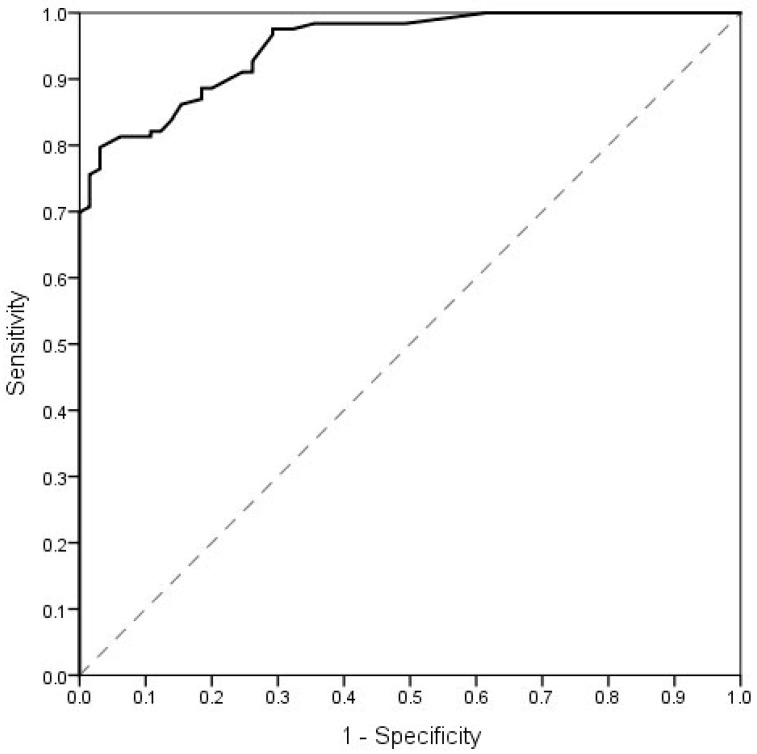
ROC curve representing the sensitivity and 1-specificity of cut-off points for the VHI-POL global score (distinction between patients with voice disorders and controls). The optimal cut-off point for VHI-POL global score was 17 points. The AUC was 0.952; *p* < 0.001. Sensitivity for the value of 17 points was 0.886 and specificity was 0.815.

**Table 1 ijerph-19-10738-t001:** Descriptive statistics for VHI items.

Number of Item	Study Group			Control Group		
	Range	*M*	*SD*	Range	*M*	*SD*
VHI_1	0–4	1.64	1.00	0–3	0.46	0.73
VHI_2	0–4	1.75	.98	0–3	0.52	0.80
VHI_3	0–4	1.99	1.20	0–3	0.62	0.88
VHI_4	0–4	2.35	0.96	0–3	0.71	0.89
VHI_5	0–4	1.68	1.16	0–4	0.48	0.94
VHI_6	0–4	1.20	1.15	0–3	0.11	0.47
VHI_7	0–4	1.81	1.24	0–2	0.11	0.40
VHI_8	0–4	1.23	1.24	0–1	0.03	0.17
VHI_9	0–4	1.41	1.15	0–2	0.18	0.43
VHI_10	0–4	2.01	1.26	0–2	0.12	0.42
VHI_11	0–4	1.13	1.16	0–1	0.06	0.24
VHI_12	0–4	1.44	1.11	0–2	0.31	0.66
VHI_13	0–4	1.87	1.02	0–2	0.32	0.64
VHI_14	0–4	2.24	1.05	0–2	0.31	0.61
VHI_15	0–4	1.59	1.23	0–2	0.12	0.45
VHI_16	0–4	1.44	1.19	0–1	0.02	0.12
VHI_17	0–4	2.12	1.10	0–2	0.37	0.60
VHI_18	0–4	1.92	1.21	0–2	0.28	0.63
VHI_19	0–4	1.07	1.15	0–2	0.03	0.25
VHI_20	0–4	1.77	1.18	0–2	0.17	0.45
VHI_21	0–4	2.06	1.18	0–3	0.38	0.74
VHI_22	0–4	0.80	1.18	0–0	0.00	0.00
VHI_23	0–4	2.52	1.23	0–2	0.28	0.63
VHI_24	0–4	1.38	1.31	0–2	0.05	0.28
VHI_25	0–4	1.12	1.23	0–1	0.02	0.12
VHI_26	0–4	1.91	1.03	0–2	0.43	0.73
VHI_27	0–4	1.55	1.22	0–2	0.34	0.67
VHI_28	0–4	1.42	1.20	0–2	0.25	0.53
VHI_29	0–4	1.27	1.17	0–2	0.08	0.32
VHI_30	0–4	1.41	1.32	0–1	0.02	0.12
Emotional subscale	0–39	15.48	10.03	0–12	1.43	2.46
Functional subscale	0–38	13.62	9.20	0–14	2.11	2.95
Physical subscale	0–37	20.00	8.00	0–15	3.61	4.49
Global score	2–107	49.11	25.67	0–31	7.16	9.02

**Table 2 ijerph-19-10738-t002:** Rho-Spearman correlations between VHI-POL scores and MDVP parameters.

MDVP Parameter	VHI-E	VHI-F	VHI-P	VHI Global
F0	0.15	0.12	0.16	0.15
Jitt	0.17	0.22 *	0.23 *	0.23 *
RAP	0.18	0.23 *	0.21 *	0.23 *
PPQ	0.18	0.23 *	0.23 *	0.24 *
sPPQ	0.12	0.18	0.20 *	0.18
vF0	0.14	0.16	0.23 *	0.19 *
Shim	0.19 *	0.24 *	0.19 *	0.22 *
APQ	0.23 *	0.26 **	0.23 *	0.25 **
sAPQ	0.18	0.22 *	0.23 *	0.22 *
vAm	0.18	0.16	0.24 *	0.20 *
NHR	−0.02	0.02	0.04	0.01
SPI	0.08	0.07	0.01	0.07

F0, average fundamental frequency; Jitt, Jitter; RAP, Relative Average Perturbation; PPQ, Pitch Perturbation Quotient; sPPQ, Smoothed Pitch Perturbation Quotient; vF0, Fundamental Frequency Coefficient Variation; Shim, % Shimmer; APQ, Amplitude Perturbation Quotient; sAPQ, Smoothed Amplitude Perturbation Quotient; vAm, Peak-to-Peak Amplitude Coefficient of Variation; NHR, Noise to Harmonic Ratio; SPI, Soft Phonation Index. * *p* < 0.05; ** *p* < 0.01.

**Table 3 ijerph-19-10738-t003:** Rho-Spearman correlations between VHI-POL scores and GRBAS (study group).

VHI Subscale and Scale	G	R	B	A	S
Emotional	0.27 **	0.25 **	0.24 **	0.34 ***	0.29 **
Functional	0.26 **	0.26 **	0.31 **	0.36 ***	0.26 **
Physical	0.35 ***	0.36 ***	0.31 **	0.26 **	0.34 **
Global score	0.30 **	0.31 **	0.29 **	0.34 **	0.32 **

G, grade; R, roughness; B, breathiness; A, asthenia; S, strain. ** *p* < 0.01; *** *p* < 0.001.

**Table 4 ijerph-19-10738-t004:** Agreement between test and retest of VHI-POL.

	Test *M* (*SD*)	Retest *M* (*SD*)	Mean Difference	*SD*	Limits of Agreement	% Agreement
VHI-E	16.00 (10.32)	17.16 (11.54)	−1.16	5.44	9.51; −11.83	95.7%
VHI-F	13.74 (9.29)	14.54 (9.70)	−0.80	4.99	8.98; −10.57	94.2%
VHI-P	20.65 (7.47)	20.80 (7.99)	−0.14	4.18	8.04; −8.33	94.2%
VHI-30 Global score	50.39 (25.38)	52.49 (27.93)	−2.10	12.65	22.68; −26.89	92.8%

VHI-E, Emotional subscale; VHI-F, Functional subscale, VHI-P, Physical subscale.

## Data Availability

Not applicable.

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
