# Peer review of "Polish Translation and Validation of the Voice Handicap Index (VHI-30)"

_ijerph, 2022, doi:10.3390/ijerph191710738_

Round 1

Reviewer 1 Report

Dear Authors,

I read your work entitled “Polish Translation and Validation of the Voice Handicap Index (VHI-30)” and here I enclose my recommendations:

General Comments

I suggest the Authors to have their text for editing of English by a professional editor or a native speaker.

Abstract

The Abstract would be good to be reconstructed. For example, this part can be omitted “…... The comparison was made of the VHI-30 scores of the patients and the controls (Mann–Whitney test; Kruskal–Wallis test), as well as calculations of criterion validity (rho-Spearman bivariate correlation), internal consistency (Cronbach’s alpha; rho-Spearman bivariate correlation), and test–retest reliability (interclass correlation coefficient) ….” and the Authors to give space to a more introductive phrase besides the 1st one “…... The aim of the study is to translate and validate a Polish version of VHI. The original 15 English-language version of VHI-30 was translated into Polish …...”.

“…. Statistically significant differences between the study and control groups were found in all items as well as in the three subscales and global score with patients scoring higher than controls….” Add here statistical results (at least the differences between control group and patients with voice disorders).

“…… There were statistically significant negative correlations between maximum phonation time and global score as well as subscales ……” Add here statistical results.

“….. Shim and APQ were correlated positively with all three subscales and the global score. There were also statistically significant correlations between VHI-30 scores and GRBAS evaluation …..” Please, do not use only abbreviations in the text and add statistical results.

“…… In the patient group there was excellent internal consistency and strong test–retest reliability for global score as well as for subscales …..” Add here statistical results (Cronbach’s alpha, etch.).

Please also provide in the abstract full number of the participants and the study’s groups.

Introduction

The introduction is pretty short and the rational of this study is written with a “basic-narrative” way. There is a partial use of the literature that has to do with the validation of VHI-30 in other languages. I suggest the Authors to include them. This will enrich the Authors “Introduction” and provide a sounder rational for the reader of their work in case of publishing (see recommended literature at the end of this review).

Methods

Please add more references in the text for the procedures you followed to characterize your sample, the diagnostic procedures, e.g. Dejonckere PH, Bradley P, Clemente P, Cornut G, Crevier-Buchman L, Friedrich G, De Heyning P, Remacle M, Woisard V. A basic protocol for functional assessment of voice pathology, especially for investigating the efficacy of (phonosurgical) treatments and evaluating new assessment techniques. Eur Arch Otorhinolaryngol, 2001, 258:77-82, is enough for this. You can see also recommended literature at the end of this review.

Result

I suggest the authors to:

1) Remove Table 1 and create a new with the comparisons of total score and VHI-30 three domains. The Authors wrote in their abstract a Mann–Whitney test and a Kruskal–Wallis test and you do not report the comparison results of all those tests. I suppose the M-W test was computed between control group and voice disordered group, on the other hand I suppose the K-W test was computed between the control group and patients’ groups with different etiology of voice disorders).

2) have under every table that has acronyms to provide the full names of them (for example it is given in Table 3 and not in Table 2)

3) to compute and report a cut-off point for VHI-30 total score and its three domains using a ROC analysis.

Discussion

The “Discussion” is written in a good way and it is connecting the results of this study with former literature. The weakness of this section is that the Authors do not use all the literature regarding the validation and cross-cultural adaptation of VHI-30, so I suggest the Authors to update this and that will make the discussion more “fruitable” (see recommended literature at the end of this review). Since I suggested ROC analysis this must be discussed as well. I believe these suggestions will make the “discussion” stronger and richer.

Suggested literature for Validation of VHI-30 and cross-cultural adaptation

Behlau, M., Dos Santos, L. D. M. A., & Oliveira, G. (2011). Cross-cultural adaptation and validation of the voice handicap index into Brazilian Portuguese. Journal of Voice25(3), 354-359.

Bonetti, A., & Bonetti, L. (2013). Cross-cultural adaptation and validation of the Voice Handicap Index into Croatian. Journal of voice27(1), 130-e7.

Gräßel, E., Hoppe, U., & Rosanowski, F. (2008). Voice Handicap Index Grading. Hno , 56 (12), 1221-1228.

Moradi, N., Pourshahbaz, A., Soltani, M., Javadipour, S., Hashemi, H., & Soltaninejad, N. (2013). Cross-cultural equivalence and evaluation of psychometric properties of voice handicap index into Persian. Journal of Voice27(2), 258-e15.

Ohlsson, A. C., & Dotevall, H. (2009). Voice handicap index in Swedish. Logopedics Phoniatrics Vocology34(2), 60-66.

Seifpanahi, S., Jalaie, S., Nikoo, M. R., & Sobhani-Rad, D. (2015). Translated Versions of Voice Handicap Index (VHI)-30 across languages: a systematic review. Iranian journal of public health44(4), 458.

Taguchi, A., Mise, K., Nishikubo, K., Hyodo, M., & Shiromoto, O. (2012). Japanese version of voice handicap index for subjective evaluation of voice disorder. Journal of Voice26(5), 668-e15.

Trinite, B., & Sokolovs, J. (2014). Adaptation and validation of the Voice Handicap Index in Latvian. Journal of Voice28(4), 452-457.

Verdonck-de Leeuw, I. M., Kuik, D. J., De Bodt, M., Guimaraes, I., Holmberg, E. B., Nawka, T., ... & Woisard, V. (2008). Validation of the voice handicap index by assessing equivalence of European translations. Folia Phoniatrica et Logopaedica60(4), 173-178.

Yun, Y. S., Kim, H., Son, Y. I., & Choi, H. S. (2008). Validation of the Korean Voice Handicap Index (K-VHI) and the clinical usefulness of Korean VHI-10. Communication Sciences & Disorders13(2), 216-241.

Zhao, E. E., Nguyen, S. A., Salvador, C. D., & O'Rourke, A. K. (2020). A meta-analysis of the association between the voice handicap index and objective voice analysis. Journal of Speech, Language, and Hearing Research63(10), 3461-3471.

Suggested literature for Methods Section

Patel, R. R., Awan, S. N., Barkmeier-Kraemer, J., Courey, M., Deliyski, D., Eadie, T., ... & Hillman, R. (2018). Recommended protocols for instrumental assessment of voice: American Speech-Language-Hearing Association expert panel to develop a protocol for instrumental assessment of vocal function. American journal of speech-language pathology27(3), 887-905.

Poburka, B. J., Patel, R. R., & Bless, D. M. (2017). Voice-vibratory assessment with laryngeal imaging (VALI) form: Reliability of rating stroboscopy and high-speed videoendoscopy. Journal of Voice31(4), 513-e1.

Suggested literature for value of the cut-off of VHI-30

Behlau, M., Madazio, G., Moreti, F., Oliveira, G., Dos Santos, L. D. M. A., Paulinelli, B. R., & Junior, E. D. B. C. (2016). Efficiency and cutoff values of self-assessment instruments on the impact of a voice problem. Journal of voice, 30(4), 506-e9.

Moradi, N., Pourshahbaz, A., Soltani, M., & Javadipour, S. (2013). Cutoff point at voice handicap index used to screen voice disorders among persian speakers. Journal of Voice, 27(1), 130-e1.

Solomon, N. P., Helou, L. B., Henry, L. R., Howard, R. S., Coppit, G., Shaha, A. R., & Stojadinovic, A. (2013). Utility of the voice handicap index as an indicator of postthyroidectomy voice dysfunction. Journal of Voice, 27(3), 348-354.

Tafiadis, D., Chronopoulos, S. K., Helidoni, M. E., Kosma, E. I., Voniati, L., Papadopoulos, P., ... & Velegrakis, G. A. (2019). Checking for voice disorders without clinical intervention: The Greek and global VHI thresholds for voice disordered patients. Scientific reports, 9(1), 1-9.

Tafiadis, D., Kosma, E. I., Chronopoulos, S. K., Voniati, L., & Ziavra, N. (2018). A preliminary receiver operating characteristic analysis on voice handicap index results of the Greek voice-disordered patients. International Journal of Otolaryngology and Head & Neck Surgery, 7(03), 98.

Coherence and Correlation of VHI-30 with acoustic measure

Dehqan, A., Yadegari, F., Scherer, R. C., & Dabirmoghadam, P. (2017). Correlation of VHI-30 to acoustic measurements across three common voice disorders. Journal of voice, 31(1), 34-40.

Tafiadis, D., Tatsis, G., Ziavra, N., & Toki, E. I. (2017). Voice data on female smokers: coherence between the voice handicap index and acoustic voice parameters. AIMS Med Sci, 4, 151-163.

Reviewer 2 Report

This is a generally very good presentation of high-quality work that is well described by the title: translation of the the Voice Handicap Index questionnaire (VHI-30) to the Polish language and validation of its applicability as a standardized tool.  The work was carried out very carefully and competently. It will, of course, be of most interest to voice clinicians and researchers dealing with Polish-speaking patients, but it could also be of interest to others interested in adapting the VHI to other languages or to others involved in similar efforts in other fields.   I think it is suitable for publication in IJERPH, possibly in essentially its present form, but I have a number of minor editorial suggestions.

In the Abstract, the term VHI-POL is used on line 27, but it has not been defined. Perhaps define it there by adding two words, so that the sentence begins, "The Polish VHI-30 (VHI-POL)...".  However, there is no need to use the acronym in the Abstract, so just use "Polish VHI-30" or "Polish VHI".

The term VHI-POL first appears in the body of the paper on line 104, again without having been defined.  I could be defined it at the end of the Introduction (Line 66).  This might read, "The aim of this study is to translate and validate a new Polish version of the VHI, which we will refer to as VHI-POL."

There is a similar problem with the use of acronyms for the subscales of the VHI-30.  I suggest introducing them in the sentence beginning toward the end of line 49: "These items are equally distributed into 3 subscales: functional (VHI-F), physical (VHI-P), and emotional (VHI-E)."

The sentence at the end of the Introduction, which I suggested modifying (see above) also appears in exactly the same form on lines 38-39 (the end of the first paragraph of the Introduction). I don't think it should appear in both places, and I think it is most appropriate at the end of the Introduction (line 66). However, I think the first paragraph needs to be expanded a bit to finish the thoughts that begin there. For example, "standardized tools should be administered" (lines 37-38), might become "standardized tools should be administered to evaluate the handicap experienced by a given patient. Handicap can be thought of as a reduction of the quality of life."  This would provide a better link to the next paragraph

Somewhere in Section 2.2 or 2.3, some information should be given about the language status of the subjects.  Presumably they were all fluent in Polish but the text should give that information and perhaps any other information about their langauage that might be relevant. Could Polish-language fluency be considered an inclusion criterion?

Line 173: "Scores were higher..." -> "For the study group, scores were highest..."

Bottom of Table 4: The acronyms for the  3 subscales should have been given above, and they should probably be used here.

Section 3.3: This paragraph uses both acronyms and descriptive names for the subscales.  Perhaps change to acronyms only?

Lines 245-2 48: I don't see how the negative correlation between VHI-POL and MPT indicates that reduced respiratory capacity and decreased MPT have a negative impact on vocal efficiency. The relationship between MPT and vocal efficieny is also true  the other way around. Where does the information about reduced respiratory capacity come from?

Reviewer 3 Report

In general terms, the manuscript is well written and presents the information in a very clear way. There are some comments that I will share with the authors, and hope they will help to improve the quality of this paper:

Abstract

1.       It is clear and well written

Introduction

2.       The authors have made a very good job presenting the importance of using questionnaires, the translations of the VHI, and acknowledging the existing Polish version of the VHI-30 and why is needed to update it?

Materials and methods

3.       Although the section 2.1. Translation from English to Polish is clear, it would be helpful to have a summary table to see the individual scores of the translators.

4.       In the sentence “mean value of three productions was taken as the representative MPT”, why did the authors select the mean value instead of the longer value (as conventional)?

5.       The subtitle “2.2. Subjects and setting” and “2.3. Participants” may overlap between them. I suggest changing the first one to “2.2. Setting” since the 2.3. Participants also includes information about the subjects

6.       Following my previous comment, I suggest moving the inclusion criteria from 2.2 to 2.3

7.       Why did the authors not include another self-administrated questionnaire to compare the results of the current translation?

Results

8.       Well presented

Discussion

9. In the sentence: “In the present study we assessed the psychometric properties of VHI-POL in measuring the effect of dysphonia on the patient’s everyday life.” There is a missing comma after study 

Reviewer 4 Report

The authors aimed to translate and validate the VHI into Polish. Their work is critical as a previous Polish translation was not rigorous and should not be used. As the VHI is an internationally accepted patient-reported outcome measure in clinical voice science, the work is significant. The work is generally well-conducted, but a few shortcomings exist in various domains, that must be addressed as follows.

1.     The patients did not rate their perceived voice disorder severity so that VHI scores can be related to mild, moderate, severe VHI handicap. See for example the original VHI paper.

2.     Method: Specify “bilingually familiar” experts. Was someone at a near-native English level? How was consensus reached after the first translations, when one translation was chosen over the other. Did everyone pick the same sentences? Was there a consensus meeting?

3.     Pilot study: Need more detailed information on the patients similar to the main study. Where exactly were the patients recruited from?

4.     Ethics statement: Need to include the name of the institution.

5.     Auditory-perceptual eval: Who performed the GRBAS? A certified SLP? The terms of the GRBAS are sometimes capitalized and sometimes not.

6.     Aerodynamic/acoustic measures: Not sure why MPT was chosen. Also, rather than using the CSL/MDVP measures a mix of selected traditional measures (jitter, shimmer, NHR/HNR) and cepstral-spectral measures (cepstral peak prominence, low to high ratio) would be more targeted. Cepstral-spectral measures liked smoothed CPP can be easily done with Praat and would allow for analysis of running speech like reading of sentences. The minimum vowel duration for analysis should be 1 sec, at least for healthy participants. For some patients it might be more difficult, but they typically can produce at least a second.

7.     Participants: Define professional voice users, e.g., singers vs. teachers. For functional dysphonia, add that you are referring to primary MTD. Sequelae post-surgery, specify the types of surgeries and what sequelae you are referring to. For example, vocal fold scar?

8.     Statistics: Not clear what the error bars in Figure 1 represent. Should always be spelled out. How many points difference indicate a clinically meaningful change (see confidence intervals and test-retest reliability)? Also, what was the exact interval for test-retest (in months, weeks)? A factor analysis is missing for the study. It would be important to verify the factor structure in Polish. Regarding the discussion of internal consistency and test-retest reliability, the results from other translations from the VHI don’t have to be listed one by one. Summarize the results instead.

9.     Terminology: Self-reported and not self-perceived; visual-perceptual and not video-perceptual; auditory-perceptual and not perceptual voice evaluations.

10.  Citations: The first sentence is missing citations. In the second paragraph, include abbreviations for all instruments listed. VRQL is the wrong abbreviation.

11.  References: Make sure to capitalize VHI throughout your references.

12.  Misc: Company is missing for EndoStrob

Round 2

Reviewer 1 Report

Dear Authors,

All the suggestion have been addressed and the manuscript has been improved. I have no further comments to do on your work.

Thank you.